# Endosonographic finding of the simultaneous depiction of bile and pancreatic ducts can predict difficult biliary cannulation on endoscopic retrograde cholangiopancreatography

**Susumu Shinoura**[1,2,3], **Akihiro Tokushige**[2,4], **Kenji Chinen**[3], **Hideki Mori**[3], **Shin Kato**[5], **Shinichiro Ueda**[2]*

1 Department of Healthcare Management, School of Psychology and Healthcare Management at Akasaka, International University of Health and Welfare, Minato, Tokyo, Japan, 2 Department of Prevention and Analysis of Cardiovascular Diseases, Graduate School of Medical and Dental Sciences, Kagoshima University, Kagoshima, Kagoshima, Japan, 3 Department of Clinical Research and Quality Management, Center of Clinical Research and Quality Management, Graduate School of Medicine, University of the Ryukyus, Nishihara, Okinawa, Japan, 4 Department of Digestive Diseases, Okinawa Chubu Hospital, Uruma, Okinawa, Japan, 5 Department of Gastroenterology and Hepatology, Faculty of Medicine and Graduate School of Medicine, Hokkaido University, Sapporo, Hokkaido, Japan

* blessyou@med.u-ryukyu.ac.jp

**Data Availability Statement:** Data cannot be shared publicly because the each patients could be imaginable from the clinical dataset including age,

## Abstract

Thus far, no curved linear array endoscopic ultrasound (CLAEUS) findings were established as predictors of difficult selective bile duct cannulation (SBDC). This study aimed to identify CLAEUS findings to predict endoscopic retrograde cholangiopancreatography (ERCP) cases with difficult SBDC. This single-center, retrospective cohort study was conducted between July 2014 and June 2017. This study included all consecutive patients who underwent CLAEUS prior to naïve ERCP. A CLAEUS finding of the simultaneous depiction of bile and pancreatic ducts at the second portion of the duodenum (D2) (simultaneous depiction) was selected as a possible predictor of difficult SBDC, and the κ values in the evaluation of inter- and intra-observer variabilities for "simultaneous depiction" were 0.65 and 0.77, respectively, with substantial correlation. Among the 986 patients who underwent ERCP, 80 patients were relevant for evaluation. Logistic regression analysis revealed strong association between "simultaneous depiction" and difficult SBDC (odds ratio 15.4, 95% confidence interval 4.2–56.0; p<0.001). Among patients who underwent CLAEUS prior to naïve ERCP, a strong correlation was observed between "simultaneous depiction" and the risk of difficult SBDC. An endoscopist can prepare for difficult SBDC by "simultaneous depiction." The finding enables pertinent planning when performing ERCP, such as setting time limits and selecting alternative devices, techniques, and skilled endoscopists, for difficult SBDC with minimal complications including post-ERCP pancreatitis. However, a future prospective study is necessary to establish the procedure algorithm for suspected difficult SBDC cases based on CLAEUS.

sex, visiting date and period of admission. Data are available from the Okinawa Chubu Hospital Ethics Committee (contact via the study protocol (H28-64) https://chubuweb.hosp.pref.okinawa.jp/contact.html) for researchers who meet the criteria for access to confidential data.

**Funding:** The author(s) received no specific funding for this work.

**Competing interests:** The authors have declared that no competing interests exist.

## Introduction

As post-endoscopic retrograde cholangiopancreatography (ERCP) pancreatitis (PEP) as a complication of ERCP may become fatal, the main concern among endoscopists is gaining access into the bile duct without unintentional cannulation at the pancreatic duct [1]. Selective bile duct cannulation (SBDC) is essential for ERCP-related procedures [2], and the endoscopist should overcome SBDC as the first step in any ERCP cases. Multiple approaches have been developed to achieve SBDC in difficult cases. Pancreatic wire-guided cannulation is the method of choice during unintentional pancreatic guidewire insertion [3,4]. Needle-knife pre-cut papillotomy and needle-knife fistulotomy are relatively safe techniques especially when performed by experienced hands [5–7]. Theoretically, the safest and least invasive way to predict difficult SBDC cases is to specify the image to delineate "a difficult SBDC case" either on a side-viewing endoscopy or a curved linear array endoscopic ultrasound (CLAEUS), by which countermeasures would be decided prior to the cannulation. With regard to side-viewing scope findings, SBDC is reported difficult when the duodenal papilla is extremely small [8], when the duodenal papilla contains a peri-ampullary diverticulum (PAD), or when the duodenal papilla has a large oral protrusion [9,10]. With regard to EUS findings, we reported in a pilot study reporting that CLAEUS findings at the second portion of the duodenum (D2), such as pressure-induced bile duct collapse at D2 and simultaneous depiction of bile and pancreatic ducts at D2, could predict difficult SBDC [11]. Given the fact that our pilot study compared only a limited number of patients, well-designed retrospective or prospective study should be performed to verify the above argument. Thus, this study aimed to find specific CLAEUS findings that would enable prediction of difficult SBDC on ERCP.

## Materials and methods

### Ethics

The institutional review board of Okinawa Chubu Hospital approved the study protocol (H28-64). Because of the retrospective nature of this study and de-identification of personal data, the board waived the need for informed consent. All procedures were performed in accordance with the ethical standards of the responsible committee on human experimentation (institutional or regional) and with the Helsinki Declaration of 1975, as revised in 2000.

### Selection and description of participants

This was a single-center observational study conducted at a tertiary referral center with 550 hospital beds and 14 ICU beds, between July 2014 and June 2017. In this study, we enrolled all patients who underwent CLAEUS prior to initial therapeutic ERCP procedures. The demographic information of the patients and related characteristics were obtained from in-hospital electronic medical records. Data collected for analysis included age, sex, ERCP (emergency or not), final diagnosis, side-viewing endoscopic findings (size of the duodenal papilla, characteristics of the duodenal papilla, and PAD), CLAEUS findings [pressure-induced bile duct collapse at D2, simultaneous depiction of bile and pancreatic ducts at D2 ("simultaneous depiction"), and common bile duct (CBD) diameter], and cases of successful and difficult SBDC according to the following definition.

Patients were divided into two groups: the straightforward group (S group), which comprised patients whose successful SBDC was completed within 20 min, and the refractory group (R group), which comprised patients whose successful SBDC was completed after more than 20 min, patients for whom the precut method was necessary, patients in whom the procedure

was performed under CLAEUS guidance (including the rendezvous technique or choledocho-duodenostomy), and patients who required a percutaneous transhepatic approach.

Patients were excluded from the study if they met any of the following criteria: 1) patients who had a tumor of the duodenal papilla or tumor invasion in the papilla (a tumor at the pancreatic head), 2) patients with surgically altered gastrointestinal or pancreatobiliary anatomy, 3) patients whose endosonographic image of a D2 did not include the duodenal papilla, bile duct, or pancreatic duct (no D2 and/or duodenal papilla images available), 4) patients whose CLAEUS were performed at outside facilities, 5) patients whose ERCP was suspended due to sudden change in physical condition including cardiopulmonary arrest, 6) patients who were scheduled for and/or actually underwent pancreatic duct cannulation without bile duct cannulation during ERCP, 7) patients who underwent CLAEUS without D2 evaluation (e.g., only for EUS fine-needle aspiration either from the esophagus or stomach, cystogastrostomy from the stomach or celiac plexus neurolysis from the stomach), 8) patients with impacted biliary stone at the duodenal papilla, and 9) patients with fluid accumulation, cysts, calcification adjacent to the ampulla or anomalous arrangement of pancreaticobiliary ducts.

## Technical information

All EUS procedures were performed with a curved linear-array echoendoscope (Olympus GF-UE260, GF-UCT240; Olympus Optical Co., Tokyo, Japan) with a universal ultrasound processor (EU-ME2; Olympus Optical Co., Tokyo, Japan). All ERCP procedures were performed with a therapeutic duodenoscope (JF240, JF260V, TJF 260V; Olympus Optical Co., Tokyo, Japan). A single-lumen cannula (ERCP catheter; MTW Endoskopie, Wesel, Germany) and a guidewire (length, 450 cm; diameter, 0.06 cm; VisiGlide2; Olympus Medical Systems, Tokyo, Japan) were used for cannulation during ERCP. Three endoscopists, all of whom had experience of performing more than 300 CLAEUS procedures and more than 700 ERCP procedures, performed or supervised the CLAEUS and ERCP.

## Definition

Successful SBDC time was defined as time from viewing the orifice of the duodenal papilla to successful selective insertion of a catheter into the bile duct. Emergency ERCP procedure was defined as positive when the ERCP was performed within 24 h after arrival at the emergency room or clinic. The final diagnoses were determined by biopsy results, clinical follow-up, and surgical pathology if available. Pressure-induced bile duct collapse at D2 was considered positive when the bile duct collapsed completely and could only be identified on the basis of the biliary duct wall (Fig 1). "Simultaneous depiction" was considered positive when the bile and pancreatic ducts were depicted simultaneously, and the visible length of each duct was >10 mm from the duodenal papilla (Fig 2). Diagnosis of PEP was made according to the Cotton's classification [12]. Diagnosis of acute cholangitis was made according to the Tokyo Guidelines 2018 (definite diagnosis of acute cholangitis: fever or laboratory data with evidence of inflammatory response+jaundice or abnormal liver function test+imaging study showing biliary dilation or imaging study showing stricture, stone, or stent) [13]. The diameter of the CBD was measured on the line drawn perpendicular to the CBD at 10–15 mm from the tip of the mucosal surface of the duodenal papilla at D2.

Thus far, as only our pilot study had asserted on the CLAEUS finding of "simultaneous depiction," validation was evaluated by assessing the inter- and intra-observer variabilities of the finding. The κ statistic is the most commonly used statistic for the evaluation of an agreement between two or more observations [14]. The degree of agreement was measured as a percentage of the total agreement using the κ-statistic to evaluate inter- and intra-observer

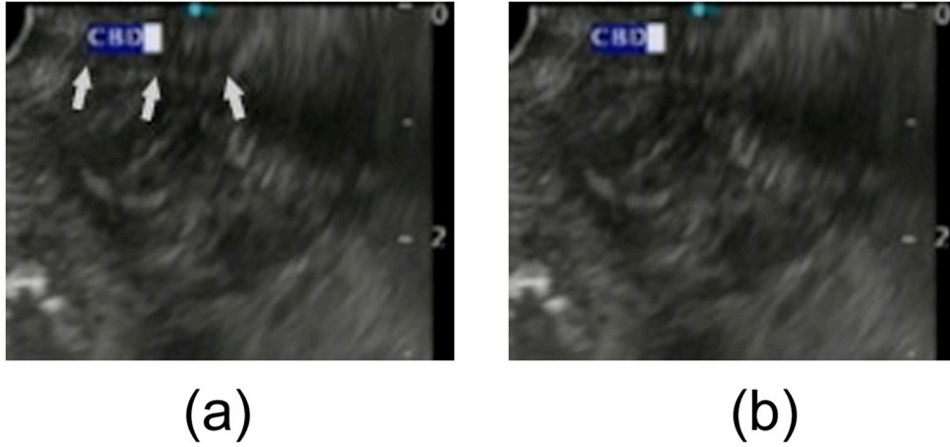

**Fig 1. <sup>a</sup>CLAEUS image of "bile duct collapse at D2", upper without arrow and lower with arrow.** Pressure-induced bile duct collapse at D2 was considered positive when the bile duct collapsed completely and could only be identified on the basis of the biliary duct wall with the pressure of the tip of the CLAEUS scope (arrow). <sup>a</sup>CLAEUS, curved linear array endoscopic ultrasound.

variabilities. A κ value of 1 means perfect agreement, whereas a κ value of 0 means agreement equivalent to chance. The quantitative classification of the κ value is shown in Table 1. To assess inter-observer variability, static images of CLAEUS were evaluated independently by two experienced endoscopists who were blinded to the procedural results and clinical outcomes. Furthermore, one of the two observers evaluated all images again at 6 months after the initial evaluation to assess intra-observer variability. The observer was blinded to the results of the initial evaluation. With regard to inter-observer variability, the CLAEUS finding of simultaneous depiction of bile and pancreatic ducts at D2 by observer 1 and observer 2 are shown in Fig 3. The κ value in the evaluation of inter-observer variability for "simultaneous depiction" was 0.65. For the intra-observer variability, the CLAEUS finding of "simultaneous depiction" by the first and second evaluations is shown in Fig 4. The κ value in the evaluation of intra-observer variability for "simultaneous depiction" was 0.77.

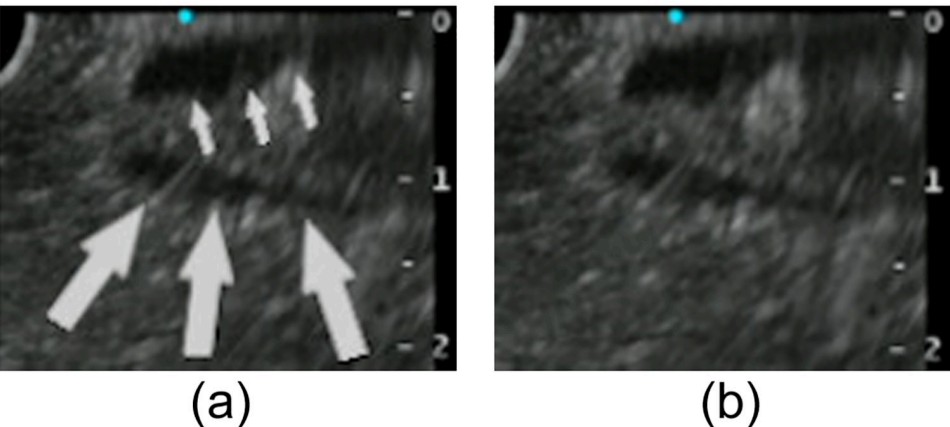

**Fig 2. <sup>a</sup>CLAEUS image of "simultaneous depiction of bile and pancreatic duct at D2".** Simultaneous depiction of the bile and pancreatic ducts at D2 was considered positive when the bile duct (small arrows) and the pancreatic duct (big arrows) were depicted simultaneously on the same axis, and the visible length of both ducts is >10 mm from the duodenal papilla. <sup>a</sup>CLAEUS, curved linear array endoscopic ultrasound.

**Table 1. Quantitative classification of kappa value.**

| kappa value | Degree of agreement |
|---|---|
| <.01 | Less than chance agreement |
| .01-.02 | Slight agreement |
| .21-.40 | Fair agreement |
| .41-.60 | Moderate agreement |
| .61-.80 | Substantial agreement |
| .81-.99 | Almost perfect agreement |

## Statistical analysis

Differences in demographic and/or clinicopathologic variables between the S group (an easy SBDC group) and R group (a difficult SBDC group) were analyzed using the chi-square and Fisher's exact tests for categorical variables and Mann-Whitney U test for continuous variables.

Based on a priori knowledge and our kappa analysis revealing validation of the CLAEUS finding of "simultaneous depiction," the following variables were incorporated into the primary multivariable model: age, sex, acute cholangitis, PAD and "simultaneous depiction."

All tests were two-tailed; $P$ values <0.05 were considered statistically significant. All statistical analyses were performed using EZR (version 1.38, http://www.jichi.ac.jp/saitama-sct/SaitamaHP.files/statmed.html).

All authors take complete responsibility for the integrity of the data and accuracy of the data analysis.

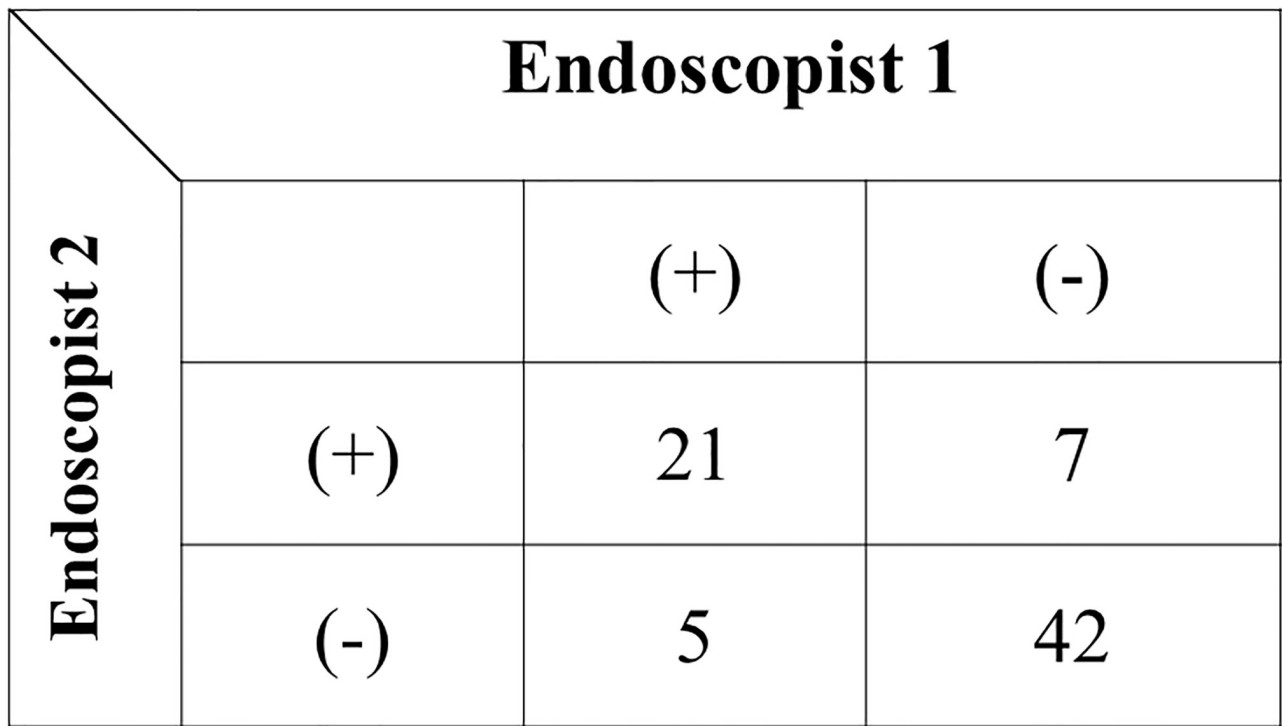

**Fig 3. Inter-observer variability.** K value = 0.65 (95%CI: 0.47–0.83). Of the five "indecisive cases" by Endoscopist 2, Endoscopist 1 labeled one case as positive and four cases as negative.

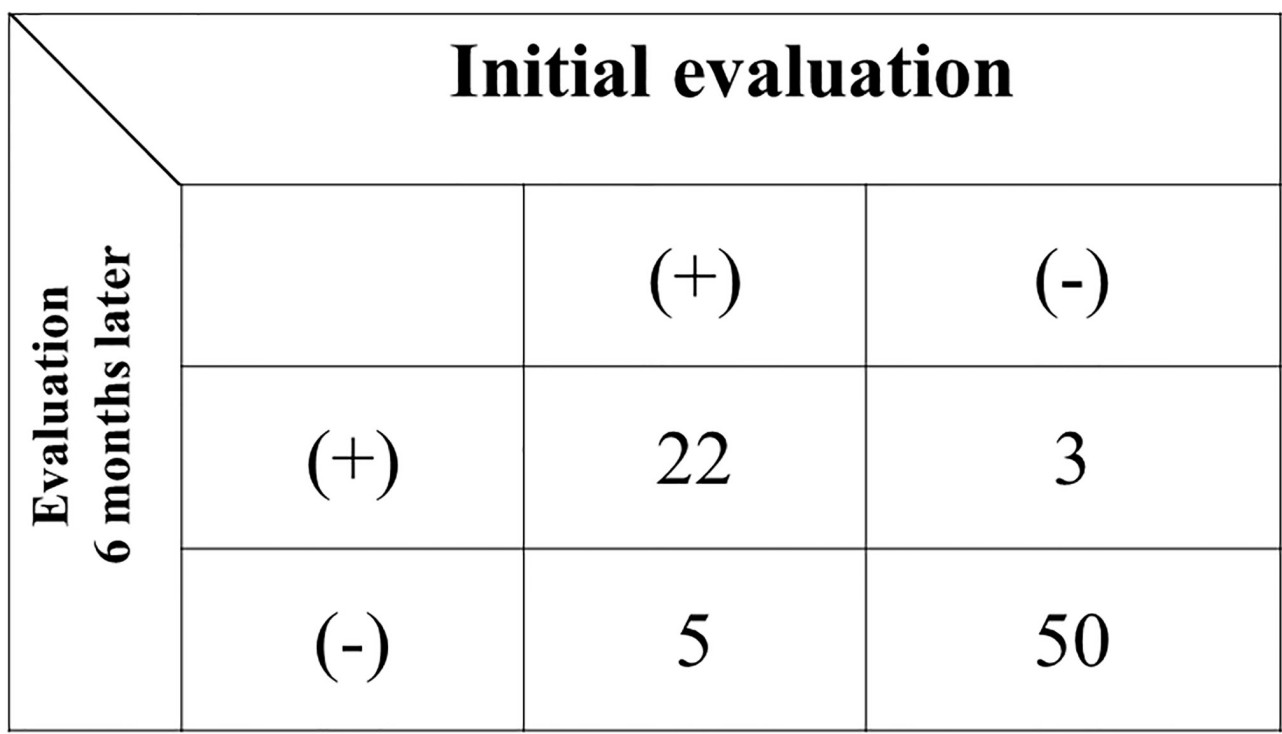

**Fig 4. Intra-observer variability.** K value = 0.77 (95%CI: 0.62–0.92).

## Results and discussion

### Results

The flow diagram of patient recruitment is shown in Fig 5. Among 986 ERCPs performed at our institution between July 1, 2014, and June 30, 2017, a total of 630 procedures were the cases with documented previous ERCP and 356 were naïve ERCP procedures. Of the above

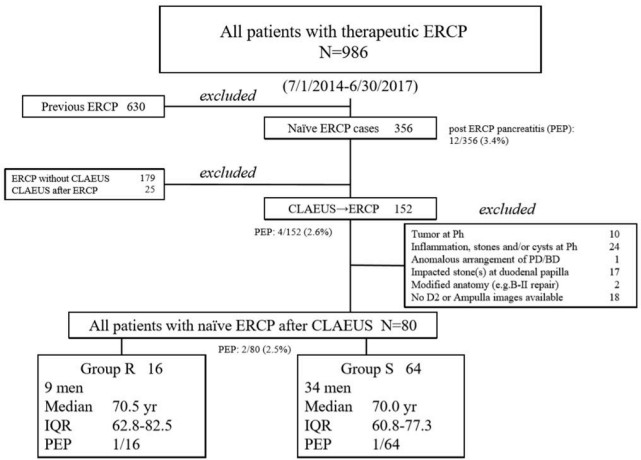

**Fig 5. Flow diagram of patient recruitment.** [a]CLAEUS, curved linear array endoscopic ultrasound; [b]ERCP, endoscopic retrograde cholangiopancreatography; [c]IQR, interquartile range. [d]Ph, pancreatic head. [e]PD, pancreatic duct. [f]BD, bile duct. [h]B-II, Billroth-II reconstruction. [i]D2, 2nd portion of duodenum.

**Table 2. Patient characteristics.**

| | All patients (N = 80) | Group R | Group S | [a]P value |
|---|---|---|---|---|
| N (%) | 80 (100) | 16 (20) | 64 (80) | |
| Age (median [[b]IQR]) | 71 [62,78] | 73 [63,83] | 70 [61,77] | 0.47 |
| Male (%) | 41 (52) | 9 (56) | 32 (50) | 0.78 |
| Emergency procedure | | | | |
| Yes (%) | 17 (21) | 6 (38) | 11 (17) | 0.09 |
| Diagnosis (%) | | | | |
| Pancreatic ductal adenocarcinoma | 3 (4) | 0 | 3 (5) | 0.99 |
| Acute pancreatitis | 4 (5) | 1 (6) | 3 (5) | 0.99 |
| Other pancreatic disorders | 3 (4) | 0 | 3 (5) | 0.99 |
| Cholangiocarcinoma | 3 (4) | 2 (13) | 1 (2) | 0.1 |
| Gallbladder cancer | 2 (3) | 0 | 2 (3) | 0.99 |
| Biliary stone and/or sludge | 31 (39) | 1 (6) | 30 (47) | 0.003 |
| Acute cholangitis without stone, sludge, mass, or obstruction | 17 (9) | 11 (69) | 6 (9) | <0.001 |
| Obstructive jaundice | 6 (8) | 1 (6) | 5 (8) | 0.99 |
| Biliary stricture | 5 (8) | 0 | 5 (8) | 0.58 |
| Other biliary disorders | 4 (5) | 0 | 4 (6) | 0.58 |
| Abnormal [c]LFT | 2 (3) | 0 | 2 (3) | 0.99 |
| Endoscopic findings (%) | | | | |
| Size of duodenal papilla, small | 33 (41) | 7 (44) | 26 (41) | 0.99 |
| Characteristics of duodenal papilla, nodulated | 6 (8) | 1 (6) | 5 (8) | 0.99 |
| Peri-ampullary diverticulum, positive | 21 (26) | 4 (25) | 17 (27) | 0.99 |
| Endosonographic findings (%) | | | | |
| Bile duct collapse | 3 (4) | 2 (13) | 1 (2) | 0.1 |
| Simultaneous [d]BD and [e]PD | 19 (24) | 11 (69) | 8 (13) | <0.001 |
| Bile duct diameter, mm | 4.1 | 3.8 | 4.1 | 0.68 |

[a]P value for group R and group S.

[b]IQR, interquartile range

[c]LFT, liver function test

[d]BD, bile duct

[e]PD, pancreatic duct

356 procedures, 204 cases were excluded and 152 were the cases with CLAEUS prior to the initial ERCPs. Of excluded 204 cases, 179 were naïve ERCP cases without CLAEUS and 25 were performed CLAEUS after naïve ERCP. Detailed information of the excluded 204 cases were listed in S1 Table. Of 152 cases who underwent CLAEUS prior to naïve ERCP, 72 cases were excluded. Detailed diagnosis of above excluded 72 cases were listed in S2 Table. Finally, a total of 80 patients who underwent CLAEUS prior to initial therapeutic ERCPs were enrolled in this study. Of the above 80 cases, 16 and 64 were categorized into the R and S groups, respectively. Characteristics of the enrolled 80 cases are summarized in Table 2. Of note, PEP was occurred 12/ 356(3.4%), 4/152(2.6%) and 2/80(2.5%) in our patient group.

Overall, the prevalence of difficult SBDC, or group R, was 20% (16/80). R group patients had less frequent diagnosis of biliary stone or sludge (R group 1 vs S group 30, P = 0.003) and more frequent diagnosis of acute cholangitis (R group 11 vs S group 6, P<0.001). No difference in age, sex, emergency procedure rate, or diagnosis, except for the above factors, was observed between the two groups. With regard to the endoscopic findings, no difference was observed in the size of the duodenal papilla, characteristics of the duodenal papilla, or PAD. For the

**Table 3. Unadjusted and adjusted odds ratios for difficult [a]SBDC in patients positive for "simultaneous depiction of bile and pancreatic ducts".**

|  | OR (95% CI) | *P* value |
|---|---|---|
| Unadjusted | 15.4 (4.2–56.0) | <0.001 |
| Adjustment 1 | 14.7 (3.9–54.1) | <0.001 |
| Adjustment 2 | 12.3 (2.5–59.6) | 0.001 |
| Adjustment 3 | 12.1 (2.5–59.4) | 0.002 |

The primary analysis (unadjusted) was performed with the logistic regression model, setting simultaneous depiction of bile and pancreatic ducts as an independent variable and difficulty in SBDC as a dependent variable.

Adjustment 1: adjusted for demographic characteristics of patients such as age and sex.

Adjustment 2: adjusted for demographic characteristics of patients as previously mentioned and presence of acute cholangitis.

Adjustment 3: adjusted for the demographic characteristics of patients as previously mentioned and presence of peri-ampullary diverticulum.

[a]SBDC, selective bile duct cannulation

CLAEUS findings, "simultaneous depiction" was observed more frequently in the R group than in the S group (R group 11 vs S group 8, *P*<0.001). CLAEUS findings of "pressure-induced bile duct collapse" and "diameter of CBD" were similar in both groups. The logistic regression analysis revealed that a CLAEUS finding of "simultaneous depiction" has a strong association with difficult SBDC (unadjusted odds ratio [OR]:15.4, 95% confidence interval [CI] 4.2–56.0; *P*<0.001). After adjusting for confounders (age, sex, acute cholangitis, and PAD), a CLAEUS finding of "simultaneous depiction" still has an association with difficulty in SBDC (Table 3). Furthermore, logistic regression analyses were performed by setting dependent variables not only with threshold of "insertion time above 20 minutes" but also above 5 minutes, 10 minutes and 15 minutes, and age, sex and "simultaneous depiction" as independent variables. Refractory cases (%) and odds ratio of the thresholds of 5 minutes, 10 minutes and 15 minutes were as follows; 47 (59%, OR:3.3, 95%CI: 0.96–11.2; *P* = 0.057), 23 (29%, OR:10.6, 95%CI: 3.2–35.2; *P*<0.001), 18 (23%, OR:23.0, 95%CI: 6–88; *P*<0.001). Except for a threshold of 5 minutes, a CLAEUS finding of "simultaneous depiction" revealed significant association with "difficult SBDC" (S3 Table).

## Discussion

### Key findings

This study mainly aimed to find the predictive findings on side-viewing endoscopy and/or CLAEUS for difficult SBDC on ERCP. Indeed, we demonstrated that the CLAEUS finding of "simultaneous depiction" has a strong correlation with the risk of difficult SBDC on ERCP even after adjustment for confounders.

Given the novelty of the CLAEUS finding of "simultaneous depiction," which, to our knowledge, was only reported in our pilot study [11], we evaluated the inter- and intra-observer variabilities to assess the feasibility and reproducibility of the CLAEUS finding. Both inter- and intra-observer variabilities in the present study indicated a substantial degree of agreement beyond chance (κ values of inter- and intra-observer variabilities were 0.65, and 0.77, respectively). Therefore, we decided to select the CLAEUS finding of "simultaneous depiction" as a candidate predictor for the risk of difficult SBDC.

The definition of difficult cannulation on ERCP varies; however, it has usually been defined according to the number of attempted cannulations and/or cannulation time, as the risk of

PEP correlates with repeated and prolonged attempted cannulation [1,15–17]. Given the above, the ideal side-viewing or CLAEUS finding to predict the risk of difficult SBDC should be obtained even without cannulating the duodenal papilla. If a certain EUS finding can play a role as a predictor for difficult SBDC, the endoscopist can prepare for a different approach such as precut or EUS-guided choledochoduodenostomy even before touching the duodenal papilla.

In performing routine CLAEUS evaluation, we perform the procedure according to the methodology given by Yamao, et al [18]. They advocated that with pertinent adjustment and rotation of the CLAEUS scope, left lobe of the liver, abdominal aorta with superior mesenteric artery and celiac artery, celiac lymph nodes, pancreatic body and tail, splenic artery and vein, spleen, left kidney and left adrenal gland, part of pancreatic head, portal vein with portal confluence and liver hilum, were scannable from the stomach. From the duodenal bulb, a gallbladder, portal vein, bile duct, common hepatic artery and pancreatic head were scannable. From the second portion of the duodenum (D2), superior mesenteric artery and vein, pancreatic head and ampulla, uncinated process of the pancreas and occasionally right kidney were scannable. In the above routine CLAEUS evaluation, D2 was the best station to evaluate the ampulla, bile duct and pancreatic duct precisely (Fig 6). Thus, we chose D2 image on CLAEUS as a candidate predictor for difficult SBDC [19].

Actually, "simultaneous depiction" is a finding subtracted from retrospective investigation of the documented CLAEUS image. Even though photodocumenting a duodenal papilla is our routine requirement, close inspection of "simultaneous depiction" is not specifically required at our institution. Therefore, "simultaneous depiction" should be an image available "without specialized or particular technique", and could be a convenient measure to use as a "predictor for difficult SBDC".

Reason why the CLAEUS finding of simultaneous depiction can predict a difficult SBDC is because the finding reflects the similarity of the anatomical structure including the angle and route of both biliary and pancreatic ducts [20], and the above similarity would easily induce wrong cannulation of the device into the pancreatic duct instead of the biliary duct. Fig 6 may help getting a vivid image of the 3-D structure at D2 and, moreover, understanding of bile and pancreatic ducts' anatomical similarity easily inducing unintentional cannulation of the pancreatic duct.

Of 152 cases with CLAEUS followed by naïve therapeutic ERCP, 72 cases were excluded. Given the substantial number of exclusion, there might be a concern for bias. Therefore, careful evaluation is required to see the reason for exclusion. On the other hand, as our current clinical study aims for searching "predictive finding for difficult SBDC", we needed to focus on "findings obtained without modification by certain lesion or condition". For above discussion, detailed diagnosis of the each case is required for evaluation. Regarding pancreatic disorders, pancreatic adenocarcinoma located at either pancreatic head or ampulla (4 cases), post pancreatobiliary surgery (2 cases), severe inflammation and/ or cystic lesion at pancreatic head (6 cases), severe fibrosis and/ or multiple calcification and/ or intraductal stones (4 cases) and stricture of main pancreatic duct adjacent to the ampulla (1 case) were the cases with pertinent reason for exclusion. Regarding hepatobiliary disorders, biliary tract cancer invading or including the ampulla (6 cases), benign stricture of ampulla with upstream dilatation (4 cases), impacted bile duct stone (17 cases) and bile duct inflammation (9 cases) and anomalous arrangement of bile and pancreatic ducts (1 cases) were the cases with pertinent reason for exclusion. However, in 18 cases, no images of duodenal ampulla were available (14 cases of obstructive jaundice, 2 cases of abnormal liver function test, 1 case of Lemmel's syndrome and 1 case of acute cholecystitis), suggesting that careful inspection of the pancreatic head and duodenal ampulla are required especially with above diagnosis from predicting the difficult SBDC

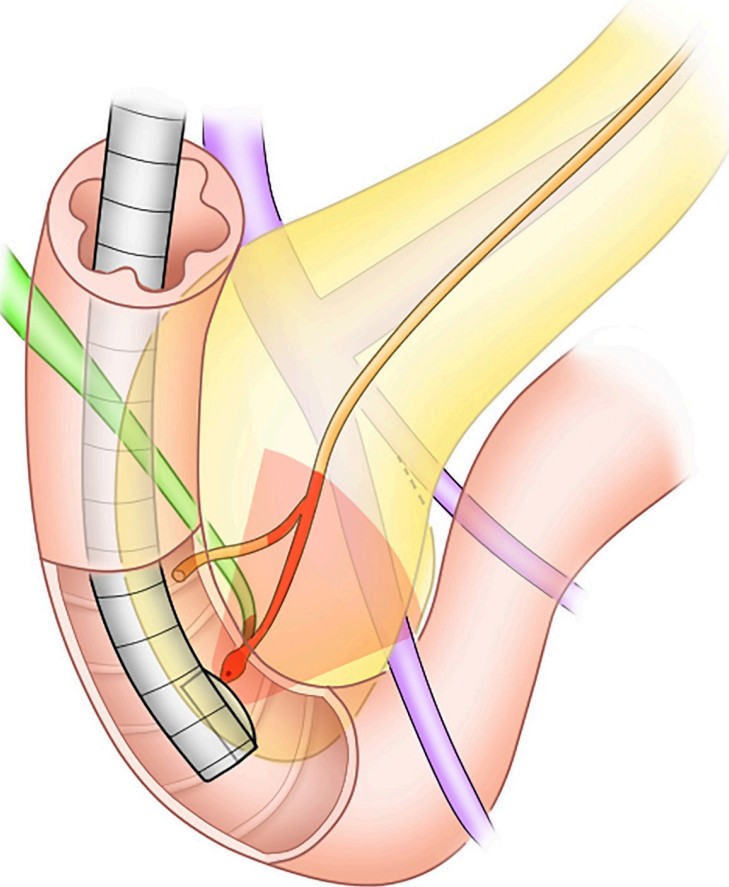

**Fig 6. A 3-dimentional view of 2ⁿᵈ portion of duodenum (D2) with a curved linear endoscopic ultrasound (CLAEUS).** Route and course of pancreatic duct and common bile duct were different even though both ducts share the same opening at duodenal papilla. The CLAEUS yields views more analogous to those obtained with transabdominal ultrasound (TAUS). The view of CLAEUS is in the same plane as the scope shaft. Red fan-shape corresponds to the view of CLAEUS at D2 focusing at duodenal papilla and pancreatic duct. When focusing on main pancreatic duct on CLAEUS, pancreatic duct and bile duct colored with dark orange are visible. Only a small portion of bile duct view is available.

standpoint (S2 Table). Especially regarding 14 cases of "obstructive jaundice", the diagnosis were made according to the finding of suspected localized caliber change of the biliary tract either on CT scan or transabdominal ultrasound with the abnormal laboratory test pattern (elevated total and direct bilirubin and gamma-GTP). Further evaluation should be planned in the future for further investigation of correlation between the finding of localized caliber change of biliary tract with "abnormal LFT pattern of obstructive jaundice" and difficulty in photodocumenting the image of duodenal papilla, including bile and pancreatic ducts.

Considering the characteristics of the R and S groups, the number of patients diagnosed with acute cholangitis and patients diagnosed with biliary stone and/or sludge were significantly higher and lower, respectively, in the R group than in the S group. With regard to acute cholangitis, it is reasonable to speculate that the inflammatory process of acute cholangitis, which was the cause of injury and edema at the duodenal papilla or bile duct, resulted in the outlet obstruction at the papilla and finally the difficult SBDC [12,21,22].

In case of biliary stone and/or sludge, the absence of biliary stone and/or sludge correlates with the smaller diameter of the biliary tract, which was one of the known risk factors for PEP [3].

Regarding cannulation time limits within which the regularly used cannulation technique is abandoned varies, such as 5 minutes [23], 10minutes [24,25], 15 minutes [26,27], 20 minutes [28] and 30minutes [29]. When we look at the distribution of cannulation time of our patient group, threshold of either more than 15 minutes or 20 minutes would be feasible, as with threshold of more than 5 minutes, 58% (47/80) of the patients should be included in the R group (S1 Fig). Furthermore, threshold of above 20 minutes was advocated and appropriately used as a threshold to perform precut by Fukatsu et al. [28]. Therefore, we decided to adopt cannulation time limit as more than 20 minutes.

Papilla contacts were not adopted as a condition for the definition of difficult SBDC, as it is difficult to differentiate between a simple, gentle touch at the ampulla without damage towards ampulla itself and/ or duct wall and a failed cannulation after manipulating the ampulla with excessive pressure, which causes bleeding, edema, or swelling at the mucosa and/ or duct wall, although both can be labelled as a "papilla contact."

The robustness of "simultaneous depiction" were evaluated with logistic regression analyses with cannulation time limit of being more than 5, 10 and 15 minutes as dependent variables. Only the threshold of "more than 5 minutes" didn't showed significant association (OR:3.3, 95%CI: 0.96–11.2, p = 0.057). Even though more-than-5-minute- threshold was adopted by ESGE for difficult SBDC, given the fact that 47 out of 80 cases (58%) were classified as R group in our study population and given the fact that our institution perform more than 500 cases of therapeutic ERCP and EUS annually and PEP complication rate of being 12/356(3.4%), 4/152 (2.6%) and 2/80(2.5%), within appropriate range compared with data from other facility such as 3.5% [30], threshold of 5 minutes in our patient group is too short and divergent and its negative association with "simultaneous depiction" didn't imply the frailty of the finding.

Several studies reported on the correlation between the anatomical structure at the bile and pancreatic duct junction and surface morphology of the duodenal papilla, and one of those reports showed that the nodular type surface of the papilla accounts for 33% of the cases, showing correlation with septal type structure, for which the achievement of SBDC is the most difficult of the four papillary structure types [31,32]. Interestingly, our study showed that the morphological and anatomical characteristics on the side-viewing scope, such as the nodular type papilla, PAD, and small-sized duodenal papilla, showed no significant difference between the R and S groups.

## Strengths and limitations

The strengths of this study are as follows: the CLAEUS finding of "simultaneous depiction of both ducts" can be obtained without touching the duodenal papilla, strongly correlates with SBDC on ERCP, and possibly decreases the risk of PEP by the earlier choice of the precut approach or interventional EUS approach or by having a skilled endoscopist perform the procedure, if the finding is positive.

Despite the novel findings, this study has limitations. First, this study had a single-center, retrospective design. Therefore, the results cannot be generalized. Second, given the relatively high number of excluded cases, it is necessary to carefully apply the result to cases under different clinical setting. Considering the retrospective study design in a single center, all CLAEUS and ERCP data within a certain period were prospectively collected and evaluated, and arbitrary inclusion or exclusion of the data was minimized. Furthermore, despite the single-center design, three experienced endoscopists who often share their cases together actually performed

or supervised the procedure; thus, side-viewing endoscopic findings such as "small papilla" or CLAEUS finding were obtained with certain objectivity and reproducibility, which was confirmed by our kappa analysis on "simultaneous depiction."

Considering the aforementioned novelty and limitation of our study, future perspectives related to the current study should (1) expand the study to multiple medical centers in Japan and possibly to different Asian countries, (2) increase the number of patients and carefully apply the result to cases under different clinical settings, and (3) include a heterogeneous population, considering age, sex, ethnicity, and background.

## Interpretations and implications

This study confirmed the results of our pilot study that the CLAEUS finding of simultaneous depiction of bile and pancreatic duct at D2 could significantly correlate with the risk of difficult SBDC on ERCP. If the CLAEUS finding of "simultaneous depiction" were used as the tool to estimate the risk of difficult SBDC in advance, a quick CLAEUS prior to ERCP could predict and allow the endoscopist to prepare for difficult ERCP cases without physically cannulating a papilla, with promising feasibility and reproducibility. By predicting the difficulty in SBDC, an endoscopist can establish pertinent planning when performing ERCP, such as setting shorter time limits and selecting alternative devices, techniques, and skilled endoscopist, to perform SBDC with minimal invasiveness. Using the CLAEUS finding of "simultaneous depiction," the endoscopist may efficiently minimize complications including PEP or unnecessary disruption of the duodenal papilla by precut in difficult SBDC cases.

## Controversies

This study suggested that the three-dimensional structure and anatomy of the bile and pancreatic ducts at the duodenal papilla is a critical factor for predicting difficult SBDC. Our results may provoke further controversy regarding the re-evaluation of "truly important findings for predicting difficult SBDC," especially the presence of a peri-ampullary diverticulum or the size of the duodenal papilla, by comparing these findings with the "simultaneous depiction of both ducts."

## Future research directions

A future prospective study is necessary to establish the treatment algorithm in cases with CLAEUS finding of simultaneous depiction of bile and pancreatic ducts. In future prospective studies, it may be feasible to evaluate whether adding other endoscopic or endosonographic findings and/ or characteristics (number of precisely and objectively defined "attempted cannulations" with cannulation time, for example) for cases with "simultaneous depiction" can contribute to not only more successful SBDC but also fewer ERCP complications including PEP.

## Supporting information

**S1 Table. Detailed diagnosis of excluded 204 cases.**
(DOCX)

**S2 Table. Characteristics of patients undergoing [a]CLAEUS-naïve [b]ERCP.**
(DOCX)

**S3 Table. Unadjusted and adjusted odds ratios for difficult ªSBDC in patients positive for "simultaneous depiction of bile and pancreatic ducts".**
(DOCX)

**S1 Fig. Distribution of "cannulation time of the patients underwent naïve ERCP after CLAEUS".**
(TIF)

## Acknowledgments

The authors would like to thank Kimitoshi Yamguchi for technical assistance with the subtraction of the data. The authors would like to thank Editage for English language editing.

## Author Contributions

**Conceptualization:** Susumu Shinoura.

**Data curation:** Susumu Shinoura, Akihiro Tokushige, Kenji Chinen, Hideki Mori, Shin Kato.

**Formal analysis:** Susumu Shinoura.

**Investigation:** Susumu Shinoura.

**Methodology:** Susumu Shinoura, Akihiro Tokushige, Kenji Chinen, Hideki Mori, Shin Kato.

**Project administration:** Susumu Shinoura, Shinichiro Ueda.

**Supervision:** Akihiro Tokushige, Shinichiro Ueda.

**Validation:** Shinichiro Ueda.

**Writing – original draft:** Susumu Shinoura.

**Writing – review & editing:** Akihiro Tokushige, Shinichiro Ueda.

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
