## [Decision Letter · Decision Letter 0]

22 Apr 2020

PONE-D-19-35600

Endosonographic finding of the simultaneous depiction of bile and pancreatic ducts can predict difficult biliary cannulation on endoscopic retrograde cholangiopancreatography

PLOS ONE

Dear Prof. Ueda,

Thank you for submitting your manuscript to PLOS ONE. After careful consideration, we feel that it has merit but does not fully meet PLOS ONE’s publication criteria as it currently stands. Therefore, we invite you to submit a revised version of the manuscript that addresses all points raised during the review process. Especially the robustness of the chosen threshold should be addressed and possibly substantiated by a cross-validation analysis.

We would appreciate receiving your revised manuscript by Jun 06 2020 11:59PM. To enhance the reproducibility of your results, we recommend that if applicable you deposit your laboratory protocols in protocols.io, where a protocol can be assigned its own identifier (DOI) such that it can be cited independently in the future. For instructions see: http://journals.plos.org/plosone/s/submission-guidelines#loc-laboratory-protocols

We look forward to receiving your revised manuscript.

Kind regards,

Hans A Kestler

Academic Editor

PLOS ONE

2. We noted in your submission details that a portion of your manuscript may have been presented or published elsewhere. ["Part of the submitted data was used for our Pilot study, which was attached as "other"with subtitle "SHINOURA Article of pilot study".] Please clarify whether this publication was peer-reviewed and formally published. If this work was previously peer-reviewed and published, in the cover letter please provide the reason that this work does not constitute dual publication and should be included in the current manuscript.

Reviewers' comments:

Reviewer's Responses to Questions

**Comments to the Author**

1. Is the manuscript technically sound, and do the data support the conclusions?

Reviewer #1: Partly

Reviewer #2: Yes

2. Has the statistical analysis been performed appropriately and rigorously? 

Reviewer #1: I Don't Know

Reviewer #2: Yes

3. Have the authors made all data underlying the findings in their manuscript fully available?

Reviewer #1: Yes

Reviewer #2: Yes

4. Is the manuscript presented in an intelligible fashion and written in standard English?

Reviewer #1: Yes

Reviewer #2: Yes

5. Review Comments to the Author

Reviewer #1: The discussed study is a single center, retrospective cohort study, including 80 patients. All patients have curved linear array endoscopic ultrasound (CLAEUS) before endoscopic retrograde cholangiopancreatography (ERCP).

The general aim of the study, was to define predictors of difficult selective bile duct cannulation (SBDC).

Hereby difficult is described as the simultaneous depiction of bile and pancreatic ducts at the second portion of the duodenum farther than 10 mm away from the papilla.

It therefore has been concluded, that if a CLAEUS is done in advance, the risk of a post-ERCP pancreatitis, as well as the risk of complications is reduced.

Limitations of the study:

I find it irritating, that the division of the patients into two groups were on the basis of the duration of the examination, and more importantly, that the used threshold, was define by the study group. Is this threshold robust? It seems to be necessary, that this threshold is varied. Furthermore additional factors in the definition of the threshold should be considered, such as the mentioned number of papilla contacts.

In Addition the number of patients were initially 356, but included in the study were only 80. This strong selected patient group I find very odd and would like a more detailed explanation of why 356 patients were included in the first place.

Reviewer #2: The work „Endosonographic finding of the simultaneous depiction of bile and pancreatic ducts can predict difficult biliary cannulation on endoscopic retrograde cholangiopancreaticography“ by Susumu Shinoura et. Al addresses an important question in the field of interventional endocopy: How is it possible to predict a difficult difficult approach to the biliary system before conventional ERCP. In the past, several attempts have been made to improve pre-therapeutic certainity about subsequent risks for patients undergoing ERCP and therefore making an early decision for possible alternative strategies like precut papillotomy or transduodenal approaches for bile drainage.

The manuscript is technically sound, the statistical analysis appears to be correct, especially the different statistical tests are chosen correctly for the distinct analyses. Both strength and limitations of the study are well described and discussed together with the current literature.

In General, I see only some minor points to be addressed:

1. As the authors state correctly the present work is based on a retrospective analysis of EUS images. This is basically discussed as a potential bias of the study. But I would recommend to further comment on an essential cause of additional errors: It is not clear whether the examiners intended to document a simultaneous depiction. There might have been a number of undocumented patients with “simultaneous depiction” but simply were not documented by the examiners. Is there standard protocol of photo-documentation in EUS, especially in D2 Position to make sure there is a high possibility that a “double depiction” is always being documented? Please comment on that more precisely.

2. In the present version of the manuscript I see some frailty with aspect to image- and graphic presentation. It might be helpful for a less specialized readership to provide a graphical overview that contains: (i) Overview of main anatomical structures, (ii) position of the echoendoscope in the duodenum and (iii) a display of “simultaneous depiction”.

I would recommend to better center the region of interest within the images of (i) bile duct collapse (Fig. 1) and simultaneous depiction (Figure 2).

3. Most parts of the text are well written in an intelligible fashion, but it might profit at a very few parts from a stylistic correction by a native speaker: E.g.: “As post-endoscopic retrograde cholangiopancreatography (ERCP) pancreatitis as a complication49 of ERCP becomes (may become) fatal…”

6. PLOS authors have the option to publish the peer review history of their article (what does this mean?). If published, this will include your full peer review and any attached files.

Reviewer #1: No

Reviewer #2: Yes: Martin Müller, MD, Dept. of Gastroenterology, Ulm University, Germany

---

## [Author Response · Author response to Decision Letter 0]

17 May 2020

May 18, 2020

Professor Joerg Heber

Editor-In-Chief 

PLOS ONE

Dear Editor:

Thank you for inviting us to submit a revised draft of our manuscript entitled “Endosonographic finding of the simultaneous depiction of bile and pancreatic ducts can predict difficult biliary cannulation on endoscopic retrograde cholangiopancreatography” to PLOS ONE. We also appreciate the time and effort you and each of the reviewers have dedicated to providing insightful feedback on ways to strengthen our paper. Thus, it is with great pleasure that we resubmit our article for further consideration. We have incorporated changes that reflect the detailed suggestions you have graciously provided. We also hope that our edits and the responses we provide below satisfactorily address all the issues and concerns you and the reviewers have noted.

To facilitate your review of our revisions, the following is a point-by-point response to the questions and comments delivered in your letter dated April 22, 2020.

Thank you very much for reviewing our manuscript and offering valuable advice.

RESPONSE

We confirmed that our manuscript met the PLOS ONE’s style requirements according to your advice. 

2. We noted in your submission details that a portion of your manuscript may have been presented or published elsewhere. ["Part of the submitted data was used for our Pilot study, which was attached as "other" with subtitle "SHINOURA Article of pilot study".] Please clarify whether this publication was peer-reviewed and formally published. If this work was previously peer-reviewed and published, in the cover letter please provide the reason that this work does not constitute dual publication and should be included in the current manuscript.

RESPONSE

The following explanation has been provided in the cover letter regarding our previous pilot study: “The pilot study (Shinoura S, Kenji C. Use of curved-linear array endoscopic ultrasonography findings to predict difficulty in biliary access during endoscopic retrograde cholangiopancreatography. EC Gastroenterology and Digestive System. 2018; 5: 306-314) was performed with a different time period from the current manuscript. Furthermore, the pilot study compared subtracted cases instead of comparing all the consecutive cases within the time period, which was different from the current manuscript. The study design of the pilot study can be described as a case control study instead of retrospective cohort study, which was also substantially different from this manuscript. The pilot study was previously peer-reviewed and published. Therefore, though the pilot study provided an inspiration, the current study does not constitute a dual publication and thus, the pilot study could be included in the current manuscript as a reference.”

RESPONSE

We have put the reason for data access restriction in the cover letter (starting at LINE 46) and researchers who meet the criteria for access to confidential data can access using the following contact details as written in the cover letter. “Data are available from the Institutional Data Access / Ethics Committee (contact via SS or xx031112@pref.okinawa.lg.jp) for researchers who meet the criteria for access to confidential data.”

4. comments from editors and reviewers

(1) Especially the robustness of the chosen threshold should be addressed and possibly substantiated by a cross-validation analysis.

RESPONSE

The definition of difficult cannulation on ERCP varies, thus, it is understandable that choosing only one threshold of “more than 20 minutes for SBDC” would be criticized as an arbitral choice. To ascertain the robustness of the finding of “simultaneous depiction”, we analyzed additional thresholds of 5, 10 and 15 minutes as not only the threshold of more than 20 minutes, but also those thresholds were advocated in previous studies (This in the Result section starting from LINE 232). Except for a threshold of more than 5 minutes, “simultaneous depiction” could significantly identify “difficult SBDC” cases (supplementary table 3). Furthermore, regarding threshold of 5 minutes, when evaluating cannulation time distribution of 80 patients (supplementary figure 1), 47 out of 80 cases (58%) were included in the “refractory group” with the threshold of more than 5 minutes. Actually, our institution perform more than 500 cases of therapeutic ERCP and EUS annually and complication rate such as post ERCP pancreatitis was 12/356 (3.4%), 4/152 (2.6%) and 2/80 (2.5%), respectively, and the complication rate was not different from those reported from other facilities, such as 3.5%. Therefore, the threshold of more than 5 minutes of cannulation time did not reflect the “true difficulty” in cannulation at our institute. Therefore, no apparent frailty was seen when more than 5-minute cannulation time was not significant (The above have been included in the Discussion section, starting at LINE 348). Unfortunately, we could not perform a cross-validation analysis due to shortage of manpower under the corona virus pandemic and restriction in accessing the data due to off-limit policy of the institution. However, the above additional analyses of the data could support the robustness of the “simultaneous depiction” to predict difficult SBDC.

(2) I find it irritating, that the division of the patients into two groups were on the basis of the duration of the examination, and more importantly, that the used threshold, was define by the study group. Is this threshold robust? It seems to be necessary, that this threshold is varied. Furthermore additional factors in the definition of the threshold should be considered, such as the mentioned number of papilla contacts.

RESPONSE

With regard to the robustness of the finding, we considered additional evaluation and discussion in 4-(1) provide answer to your area of concern. Furthermore, the following discussion has been added in the manuscript (in the DISCUSSION section starting from LINE 356): “Papilla contacts were not adopted as a condition for the definition of difficult SBDC, as it is difficult to differentiate the simple, gentle touch at the ampulla. In addition, failed cannulation after manipulating the ampulla with excessive pressure causes bleeding, edema, or swelling at the mucosa and or duct wall." 

(3) In Addition the number of patients were initially 356, but included in the study were only 80. This strong selected patient group I find very odd and would like a more detailed explanation of why 356 patients were included in the first place.

RESPONSE

Of the 356 naïve ERCP cases, the cases with and without CLAEUS being performed after naïve ERCP were 25 and 179, respectively. We have indicated this in the flowchart (Figure 5) and table (supplementary table 1) in the result (This is indicated in the RESULT section starting from LINE 194) and further commented on in the discussion section (starting from LINE 313). Regarding 152 cases with CLAEUS followed by naïve therapeutic ERCP, 72 cases were excluded. Additional list was also added to describe the diagnosis and detailed information of each excluded case (supplementary table 2). There were 18 cases without available image of the duodenal papilla and or D2 image. This was commented on in the Result section (starting from LINE 331): “Especially with regard to the 14 cases of “obstructive jaundice”, the diagnoses were made according to the finding of suspected localized caliber change of the biliary tract either on CT scan or trans abdominal ultrasound with the abnormal laboratory test pattern (elevated total and direct bilirubin and gamma-GTP). Further evaluation should be planned in the future for further investigation of the correlation between the finding of localized caliber change of biliary tract with “abnormal LFT pattern of obstructive jaundice” and difficulty in photo-documenting the image of duodenal papilla, including bile and pancreatic ducts.”

(4) As the authors state correctly the present work is based on a retrospective analysis of EUS images. This is basically discussed as a potential bias of the study. But I would recommend to further comment on an essential cause of additional errors: It is not clear whether the examiners intended to document a simultaneous depiction. There might have been a number of undocumented patients with “simultaneous depiction” but simply were not documented by the examiners. Is there standard protocol of photo-documentation in EUS, especially in D2 Position to make sure there is a high possibility that a “double depiction” is always being documented? Please comment on that more precisely.

RESPONSE

We did not intentionally evaluate the “simultaneous depiction” at D2. However, it is our routine to visualize and photo-document the duodenal papilla and both ducts from the papilla. We consider the above comment further reinforce the usefulness of our finding as a tool for prediction of difficult SBDC. We have reflected this in the Discussion section, starting from LINE 292.

Our response here is similar to that provided for reviewer comment 4-(3): “14 out of 18 cases with unavailable photo-documentation of D2 were the cases with obstructive jaundice. The cases were diagnosed with abnormality of the biliary tract caliber and abnormal liver function test (supplementary table 2). We could not clarify the reason why D2 was difficult to visualize in those cases, and we commented that future studies will be required. 

We did not generalize our routine step by step methodology of the routine CLAEUS. We described it in detail in the Discussion section (starting from LINE 280). In addition to above, we specifically added the illustration (figure 6) to help the understanding of the readers that D2 inspection is important in the evaluation of the duodenal papilla, bile duct, and pancreatic duct structure. We added further explanation on the correlation of “simultaneous depiction” and difficult SBDC in the Discussion section (starting from LINE 310).

(5)In the present version of the manuscript I see some frailty with aspect to image- and graphic presentation. It might be helpful for a less specialized readership to provide a graphical overview that contains: (i) Overview of main anatomical structures, (ii) position of the echoendoscope in the duodenum and (iii) a display of “simultaneous depiction”.

I would recommend to better center the region of interest within the images of (i) bile duct collapse (Fig. 1) and simultaneous depiction (Figure 2).

RESPONSE

For a less specialized readership, we have added an illustration of the D2 (figure 6) to help the understanding of the graphical overview, with the main anatomical structures, position of the CLAEUS scope at D2, and the view which could be gained from the position. This comment has been added in the Discussion section (starting from LINE 280).

Regarding Figures 1 and 2, we did our best to center the region of interest of the (i) bile duct collapse (Fig. 1) and the simultaneous depiction. 

(6) it might profit at a very few parts from a stylistic correction by a native speaker: E.g.: “As post-endoscopic retrograde cholangiopancreatography (ERCP) pancreatitis as a complication49 of ERCP becomes (may become) fatal…”

RESPONSE

As the reviewer suggested, this part has been revised by a native English speaker. The sentence in the Introduction section, starting from LINE 47 now reads as follows: “Post-endoscopic retrograde cholangiopancreatography (ERCP) pancreatitis (PEP) as a complication of ERCP may become fatal……”

Again, thank you for giving us the opportunity to strengthen our manuscript with your valuable comments and queries. We have significantly improved our manuscript by incorporating your feedback and we hope that these revisions makes our submission acceptable and suitable for publication in your esteemed journal.

Sincerely,

Shinichiro Ueda, MD., PhD.

Department of Clinical Research and Quality Management

University of the Ryukyus Graduate School of Medicine

207 Uehara, Nishihara-cho, Okinawa 903-0215, Japan

Tel: +81-98-895-1195 

Fax: +81-98-895-1447

E-mail: blessyou@med.u-ryukyu.ac.jp

---

## [Decision Letter · Decision Letter 1]

23 Jun 2020

Endosonographic finding of the simultaneous depiction of bile and pancreatic ducts can predict difficult biliary cannulation on endoscopic retrograde cholangiopancreatography

PONE-D-19-35600R1

Dear Dr. Ueda,

We’re pleased to inform you that your manuscript has been judged scientifically suitable for publication and will be formally accepted for publication once it meets all outstanding technical requirements.

Kind regards,

Hans A Kestler

Academic Editor

PLOS ONE

Additional Editor Comments (optional):

Reviewers' comments:

Reviewer's Responses to Questions

**Comments to the Author**

1. If the authors have adequately addressed your comments raised in a previous round of review and you feel that this manuscript is now acceptable for publication, you may indicate that here to bypass the “Comments to the Author” section, enter your conflict of interest statement in the “Confidential to Editor” section, and submit your "Accept" recommendation.

Reviewer #1: All comments have been addressed

Reviewer #2: All comments have been addressed

2. Is the manuscript technically sound, and do the data support the conclusions?

Reviewer #1: Yes

Reviewer #2: Yes

3. Has the statistical analysis been performed appropriately and rigorously? 

Reviewer #1: N/A

Reviewer #2: Yes

4. Have the authors made all data underlying the findings in their manuscript fully available?

Reviewer #1: Yes

Reviewer #2: Yes

5. Is the manuscript presented in an intelligible fashion and written in standard English?

Reviewer #1: Yes

Reviewer #2: Yes

6. Review Comments to the Author

Reviewer #1: The comments were addressed and precisely differentiated in the revised version. They have added helpful explanations for the reader.

Reviewer #2: The authors have fully adressed all of my comments.

(Only due to formatting reasons) the Resolution of the Pictures is quite low, this should be corrected within the final version of the manuscript.

7. PLOS authors have the option to publish the peer review history of their article (what does this mean?). If published, this will include your full peer review and any attached files.

Reviewer #1: No

Reviewer #2: No

---

## [Editor Report · Acceptance letter]

25 Jun 2020

PONE-D-19-35600R1 

Endosonographic finding of the simultaneous depiction of bile and pancreatic ducts can predict difficult biliary cannulation on endoscopic retrograde cholangiopancreatography 

Dear Dr. Ueda:

I'm pleased to inform you that your manuscript has been deemed suitable for publication in PLOS ONE. Congratulations! Your manuscript is now with our production department. 

Kind regards, 

on behalf of

Prof. Hans A Kestler 

Academic Editor

PLOS ONE